# Assessment of Satisfaction, Compliance and Side Effects among Long-Term Orthokeratology Wearers

**DOI:** 10.3390/jcm11144126

**Published:** 2022-07-15

**Authors:** Shang-Yen Wu, Jen-Hung Wang, Cheng-Jen Chiu

**Affiliations:** 1Department of Ophthalmology, Hualien Tzu Chi Hospital, Buddhist Tzu Chi Medical Foundation, Hualien 970, Taiwan; sun90602@gmail.com; 2Department of Medical Research, Hualien Tzu Chi Hospital, Buddhist Tzu Chi Medical Foundation, Hualien 970, Taiwan; jenhungwang2011@gmail.com; 3Department of Ophthalmology and Visual Science, Tzu Chi University, Hualien 970, Taiwan

**Keywords:** satisfaction, compliance, survey, myopia, orthokeratology (Ortho-k)

## Abstract

Purpose: This study aims to assess the satisfaction, compliance, and side effects among the long-term orthokeratology (Ortho-K) users in a tertiary hospital in Taiwan and analyze the side effects and related risk factors. Methods: Children and their guardians were assessed using a structured and validated questionnaire inquiring about background information, wear and care behaviors, daily activities, satisfaction, and related concerns. Clinical information, including refractive data and side effects, was obtained through patient medical files. Results: Three hundred and five school-aged patients were enrolled, and the average age was 13.13 ± 3.39 years, with an average wearing period of 17.1 ± 8.1 months. Over 83% of the subjects had clear daytime vision all day, around 88% felt satisfied or very satisfied with the results, and 98% exhibited a willingness to continue wearing the Ortho-K lenses. Most guardians (83%) were pleased with the controlling effect of myopic progression. Initial spherical equivalent and regular cleaning of the lens protein significantly correlated with clear day vision. Wearing >6 days/week correlated with less risk of lens binding. Based on the questionnaire, the main reasons for using Ortho-K were effectiveness, safety, and practicality, while the major concerns were discomfort, harmful to the eyes, and no effect. Conclusion: With a comprehensive care program from practitioners and good compliance of users, Ortho-K could be the most effective and satisfactory option for myopic children in Taiwan.

## 1. Introduction

Myopia is one of the leading eye diseases that exerts a great impact on global health as it could cause visual impairment and affect daily life [1]. Of note, high myopia (<−6.00 diopters) is associated with various serious complications like cataracts, macular degeneration, choroidal neovascularization, glaucoma, and retinal detachment [1,2,3]. A recent nationwide survey in Taiwan reported >85% prevalence of myopia in high school students, with 30% having high myopia [4]. High expectations related to educational outcomes, extensive use of after-school tutorials, and heavy use of electronic devices among children correlates with a high incidence of myopia [4]. Several studies also reported a correlation between increasing age accompanied with a decline in time spent outdoors. [5,6,7,8]. The modern East Asian education system and the popularity of portable electronic devices could be the major challenges for myopia control [9]. Hence, there is an essential and urgent need to effectively prevent myopia onset and progression.

Orthokeratology (Ortho-K) is one of the common methods to control myopia in Asia, and the number of users and related studies have grown in recent years [10,11]. A global survey of myopia management attitudes and strategies among eye care practitioners reported that Ortho-K was considered the most effective method of myopia control [12]. In 2008, the Taiwan Food and Drug Administration approved Ortho-K lenses. With the benefit of clear day vision and efficacy in myopia control, the overall extent of Ortho-K fitting has risen in Taiwan since then [13].

Nevertheless, Ortho-K treatment has several concerns. First, an impractical outlook and misinterpretation of Ortho-K from parents could affect the efficacy and safety of the treatment. Secondly, a lack of proper cleaning and wearing procedures could cause side effects, such as conjunctivitis and microbial keratitis [14]. Thirdly, although most wearers have good daytime vision, the factors affecting the efficacy and discomfort symptoms of Ortho-K users have rarely been explored. Finally, although Ortho-K treatment has a 40–60% mean reduction in the axial length compared with spectacles use [15,16], this effect could be affected by the wearing behaviors and time spent on near-work activities [9]. Thus, investigating the subjective feelings, compliance, and daily activity of users during lens wear could help practitioners offer specific support and guidance for Ortho-K wearers and their parents.

Hence, through a questionnaire, this study aims to examine the clinical record of long-term Ortho-K users from a tertiary hospital in Taiwan, including the wearing behaviors, time spent at near-work versus outdoor activities, comfort levels, visual outcome, and satisfaction. Moreover, this study investigates the correlation between the side effects retrieved from patients’ medical files and risk factors based on the questionnaire results. This study will be applicable for an integrated understanding of real-world practice in Ortho-K use through clinical and user aspects.

## 2. Materials and Methods

### 2.1. Subjects

We surveyed 478 myopic children who were 6 to 18 years old at follow-up visits at Hualien Tzu-Chi General Hospital between December 2020 and November 2021. All subjects were regular patients and had the best-corrected visual acuity of 20/25 or above at their first visit, with no ocular disease or contra-indications for wearing Ortho-K lenses. Only patients who had used a pair of Ortho-K lenses that were used for at least 6 months and were still on Ortho-K treatment during the surveyed period were eligible for this study. This study adhered to the tenets of the Declaration of Helsinki and was approved by the Institutional Ethical Committee Review Board of Hualien Tzu-Chi General Hospital. The purpose and process of the study were carefully explained to all subjects and their guardians before obtaining the signed consent form. All subjects and their guardians were assessed using a structured questionnaire with an adequate explanation by our assistant. A total of 305 children met the inclusion criteria and agreed to participate in the study. Furthermore, relevant clinical information, such as pre-treatment refractive error and keratometry readings, duration of Ortho-K treatment, lens design used, visual acuity, and incidence of side effects, was retrieved and confirmed from patients’ medical files and confirmed by the same doctor (C.-J. Chiu).

### 2.2. Lenses and Handling of Lenses

The Ortho-K lenses are four-zone reverse-geometry lenses (Euclid Systems Orthokeratology; Euclid System, Herndon, VA, USA) manufactured using Boston Equalens II material, with a nominal Dk of 127 × 10^−11^ (cm^2^/s; mL O_2_/mL × mm Hg). The same practitioner (C.-J. Chiu) fitted all Ok lenses using the standardized fitting criteria. All subjects were recommended to wear their lenses for at least 7 h every night while sleeping. In addition, they were asked to clean their lenses using a povidone iodine–based (iodopovidone-based) rigid lens disinfecting solution or a multipurpose solution with lens rubbing. Furthermore, they were asked to perform biweekly lens protein removal and replacement of the lens case and holder every 3 months. The replacement of new lenses was suggested every 2 years.

### 2.3. Follow-Up Visit

The average clinical follow-up interval was 2–3 months. In follow-up examinations, unaided vision, best-corrected visual acuity (BCVA), and findings of a slit-lamp examination of each visit were recorded. Furthermore, both the lenses and lens cases were inspected to assess lens condition. Corneal topography examination and axial length measurement were performed every 3 and 6 months, respectively. All the examinations were completed before 11 a.m. within 2 h after the contact lens removal.

### 2.4. Questionnaire

We designed a multiple-index questionnaire after adopting the questionnaires from previous studies [4,10,17,18] and the Quality-of-Life Impact of Refractive Correction (QIRC) [19]. The structured questionnaire comprised the following six domains: (i) demographic and background information; (ii) time for near-work and outdoor activities during weekdays and weekends; (iii) wearing behavior and follow-up visit; (iv) satisfaction with Ortho-K and willingness to continue wearing; (v) major discomfort symptoms and grades of severity; and (vi) reasons and major concerns for using Ortho-K.

While children themselves answered questions about satisfaction and discomfort with the Ortho-K treatment, questions about contact lens–related symptoms, time for near-work and outdoor activities, and lens handling were answered with help from the accompanying guardian. For each symptom occurring in the previous 6 months, subjects were asked to grade the frequency/severity on a 5-point scale. Furthermore, the reasons and major concerns for using Ortho-K were answered by their parent/guardian via picking the top three choices.

### 2.5. Statistical Analysis

All statistical analyses were performed using the SPSS 21.0 software package (IBM Inc., Chicago, IL, USA). Data are expressed as frequencies, proportions, or means ± standard deviations, depending on the characteristics of each item. We used a *χ*^2^ test or Fisher’s exact test to assess the correlation between two categorical variables. Using an independent-sample t-test, we compared the means of continuous variables between two groups. In addition, we used logistic regression to evaluate the correlation between risk factors and side effects. Furthermore, crude and adjusted odds ratios (ORs) and 95% confidence intervals were calculated. In this study, a two-tailed *p* = 0.05 was considered statistically significant.

## 3. Results

### 3.1. Demographics and Background Information

A total of 305 Ortho-K users were enrolled (155 males and 150 females; average age: 13.15 years). Table 1 summarizes the subjects’ demographic data. Around 38% of them were primary school (<12 years old) students, 37% were junior high (12 to 15 years old) school students, and 25% were senior high school (>15 years old) students. The average wearing period of Ortho-K was 17.1 ± 8.1 months. Compared with males, females had a higher initial spherical refractive error, cylinder power, and keratometry readings, but shorter axial length.

Our subjects’ parents had a high education level; the education level of around 80% was college and above. Most parents (54–63%) were parents of subjects with moderate (−6D to −3D) to high myopia (<−6D). The major reason for adopting Ortho-K treatment was a recommendation from relatives and friends, followed by a recommendation from doctors.

Table 2 lists the background information of Ortho-K users. Around 61% reported having myopia for at least 3 years. Before commencing Ortho-K, 72% wore spectacles, 10% used soft contact lenses, and 20% used different doses of atropine eyedrops to control myopia. The main reasons for wearing Ortho-K included: “an effective myopia control treatment” (76%), followed by “inconvenient to wear glasses” (65%), and “only need to wear at night” (48%). The major reasons for choosing the hospital as the fitting place were: “the professionalism of the doctor” (55%), “recommended by relatives or friends” (53%), and “well-planned fitting and regular follow-up” (50%).

### 3.2. Insertion Behavior and Follow-Up Visits

In this study, 66% of Ortho-K users wore and cleaned the lenses themselves. Older subjects reported a higher rate of the self-insertion option. Around 89% of wearers regularly visited the clinic every 2–3 months. As subjects grew older, a growing number wore and cleaned the lenses by themselves. However, the regular follow-up rate decreased from 97% in primary school users to 77% in senior high school users (Table 3).

Approximately 93% of users wore the lens for >6 h each day, and around 84% wore them >6 days/week. Most subjects (90%) replaced the lens case and holder orderly. More than half of the wearers (52%) cleaned their lenses with povidone iodine–based disinfecting solution and performed protein removal regularly, and 85% of them replaced their lenses on schedule if recommended. About 25% of users had lost or damaged their lenses during their insertion period (Table 4).

### 3.3. Time for Near-Work and Outdoor Activities during Weekdays and Weekends

Table 5 summarizes the daily activities of Ortho-K users. The junior and senior high school students spent more time on near-work activities during weekdays and weekends than elementary school students. The proportion of spending >8 h in near-work activities increased with age. Among subjects aged >12 years, 40%–62% spent >8 h in near-work activities during weekdays.

Generally, subjects aged >12 years spent more time using electronic devices during weekdays and weekends than those aged <12 years. Time spent using electronic devices increased with age, especially in female subjects aged >15 years.

Furthermore, time spent on outdoor activities during weekdays and weekends decreased with age in general. Nevertheless, we observed gender differences among subjects aged >15 years; while male subjects spent more time on outdoor activities during weekdays and weekends, around 69% of female subjects spent <1 h on outdoor activities during weekends.

### 3.4. Satisfaction with Ortho-K

Table 4 and Table 6 summarize the survey of satisfaction. Over 83% of subjects had clear vision all day. Approximately 66% of users felt it was easy or very easy to wear the Ortho-K lenses, and 83% of the subjects and guardians were satisfied with the controlling effect of the axial length elongation. Generally, around 88% of subjects felt satisfied or very satisfied, and most of them (98%) reported a willingness to continue wearing the Ortho-K lenses. Nonetheless, around 65% of them felt that the Ortho-K lenses were a little expensive.

### 3.5. Factors Affecting Clear Day Vision and Side Effects

Although all subjects had different periods of commencement, most of them (89%) had regular visits and their unaided vision was measured at each follow-up visit. After reviewing the clinical records, we found that 248 subjects (81.3%) had unaided vision greater than or equal to 20/25, 25 subjects (8.1%) had unaided vision greater than or equal to 20/30 and less than 20/25, and 23 subjects (7.6%) had unaided vision greater than or equal to 20/40 and less than 20/30 at each visit. Only 9 subjects (3%) had fair unaided vision (<20/40) and 6 children need wearing glasses during the daytime. These results were consistent with the subjective responses (clear day vision) from the questionnaires. Using logistic regression, we analyzed the factors related to clear day vision. Co-variates included age, sex, initial spherical equivalent (SE), initial average keratometry, initial astigmatism, toric lenses use or not, inserting lenses by themselves or not, wearing time per day, number of insertion days per week, regular cleaning of lens protein or not, cleaning methods, and frequency of clinic visit and duration of Ortho-K treatment (Table 7).

According to the results in Table 7, clear day vision was significantly correlated with an older age (>15 years; OR 0.22, *p* < 0.001), less initial SE (OR 1.54, *p* < 0.001), and regular follow-up (OR 3.05, *p* = 0.006). Moreover, subjects who wore the Ortho-K lenses with others’ help was correlated with clear day vision (OR 2.05, *p* = 0.050) compared with those inserting by themselves. However, after adjusting for other co-variates, only initial SE and regular cleaning of lens protein significantly correlated with clear day vision (OR 1.60 and 2.53, *p* < 0.001 and 0.038, respectively).

According to the subjects’ reports and their medical records, lens binding (34.8%), lens decentration (15.4%), and punctate keratitis (7.9%) were leading side effects reported within the previous 6 months. We also analyzed the factors affecting these side effects. Remarkably, the insertion days were significantly correlated with lens binding. Wearing the Ortho-K lenses ≥6 days/week was correlated with less risk of lens binding compared with wearing them ≤5 days/week (OR 0.46, *p* = 0.044). Moreover, initial SE was marginally significantly correlated with lens binding (OR 0.90, *p* = 0.075); however, the significance changed to non-significance after adjusting for other co-variates. Regarding punctate keratitis and lens decentration, we found no significant co-variates (Table 7).

### 3.6. Discomfort Symptoms

Figure 1 shows the prevalence and severity of discomfort symptoms. The top three discomfort symptoms were secretion (37%), lens binding (35%), and itching (32%; Figure 1a). When users were asked to grade the severity of each discomfort symptom from 1 to 5, the results revealed that the severities of these discomfort symptoms were 1.46–1.95, which represented mild severity. Itching had the highest severity (1.95), followed by excessive blinking (1.93) and tearing (1.81; Figure 1b).

### 3.7. Reasons and Major Concerns for Using Ortho-K

Figure 2 shows the reasons and concerns of Ortho-K use from the parents/guardian. The top three reasons for using Ortho-K were effectiveness (95%), safety (73%), and practicality (65%). Most parents opted for Ortho-K because of its effectiveness, with 64% ranking this as the key reason (Figure 2a). The top three major concerns for using Ortho-K were discomfort (86%), harmful to the eyes (80%), and effectiveness (76%). Although 86% of parents chose discomfort as the main concern for using Ortho-K, around 48% ranked “harmful to the eyes” as the main concern (Figure 2b).

## 4. Discussion

To the best of our knowledge, this is the first large-sample size study that provides abundant information about the practice of Ortho-K use in a tertiary hospital in Taiwan. We assessed the correlation between the clinical results and side effects during Ortho-K treatment. The five major findings of this study are as follows: (i) >83% of subjects had clear daytime vision all day, and around 88% felt satisfied or very satisfied with the results and 98% exhibited a willingness to continue wearing the Ortho-K lenses. Most parents (83%) were also pleased with myopia control. (ii) Initial SE and regular cleaning of lens protein was significantly correlated with clear day vision. (iii) Most subjects had good compliance, but the regular follow-up rate decreased from 97% in primary school users to 77% in senior high school users. (iv) Younger subjects spent more time on outdoor activities during weekdays and weekends, and time spent using electronic devices increased with age, especially among older females. (v) The main reasons for using Ortho-K were effectiveness, safety, and practicality, while the major concerns were discomfort, harmful to the eyes, and effectiveness.

The refraction clinic of Hualien Tzu Chi Hospital has been providing an Ortho-K service since 2013 and has established a comprehensive protocol for parents or users to enhance their compliance. Before lens prescription, an ophthalmologist has face-to-face communication with patients/parents about the relevant risks and benefits of the Ortho-K. The complete examination, including refraction status, corneal topography, and axial length measurement, and trial lens fitting are performed before lens ordering. Furthermore, we implemented a care system to send reminders for return visits and when to replace accessories and lenses. After starting lens wearing, parents or users were invited to participate in a closed social media (Facebook) group (“Explore the new vision”) for two-way communication. In our survey, 83% of subjects had clear day vision, and 95% opted for the Ortho-K because of its effectiveness. Our study also showed good compliance and was consistent with previous studies [9,13,20]. Notably, 83% of parents were satisfied with the myopia control effect. Chang et al. suggested that parental compliance with scheduled follow-up visits correlated with whether they were informed of axial length changes during the consultation [9]. Hence, a complete fitting and care program for Ortho-K is essential and could improve compliance among Ortho-K users.

The demographic information showed most wearers were primary school children with a similar gender distribution. Female subjects reported higher initial refractive error and keratometry readings than male subjects, but they had a shorter axial length. Most parents were highly educated and had moderate to high myopia. They learned about Ortho-K through word-of-mouth or from ophthalmologists. A previous study indicated that the acceptance rate for Ortho-K commencement was significantly correlated with high myopia in one of the parents, high education level, and knowledge of Ortho-K for myopia control [17]. In Taiwan, Ortho-K is a selective treatment for myopia that is not covered by insurance. Parents with high education could search for more information related to the Ortho-K lenses and discuss it with their ophthalmologist. In our study, parents were also concerned about the disadvantages of Ortho-K, like it being “harmful to the eyes”, and was consistent with a previous study [21]. Thus, parents and eye care practitioners should be diligent in regular aftercare to ensure a safe strategy and minimize the risks involved.

In our clinic, we found that the duration between diagnosis of myopia and seeking therapy has been decreasing annually, perhaps because of the increasing public attention paid to myopia and awareness of myopia control options in parents of young children. Parents of their children receiving Ortho-K might inform those of other myopic children and help them to make a decision [9]. As the progression of childhood myopia slows with age, younger children could benefit more from wearing the Ortho-K lenses for myopia control than older children. Hence, early detection and commencement of control could reduce the risk of high myopia.

In this study, 77.5% and 50.1% of Ortho-K users spent >6 h in near-work activities during weekdays and weekends, respectively. The proportion of spending time on electronic devices increased with age, and the time for outdoor activities decreased with age in general; this aligns with a recent study in Taiwan [4]. Previous studies indicated that >3 h of near-work activities per day could be the main risk factor for myopia [4,22]. Meanwhile, electronic devices have been integrated into the everyday life of schoolchildren. Moreover, the modern East Asian education system and the popularity of portable electronic devices could be responsible for the early high prevalence of myopia (25.41%) and high myopia (24.16%) in 7-year-old Taiwanese children [22].

Furthermore, during the COVID-19 pandemic, students were confined to homes, markedly increasing their indoor activities and screen time and, thus, decreasing their outdoor activities. This is possibly correlated with a significant myopic shift for children [23]. Parents should be aware that prolonged near-work activities and overuse of electronic devices is correlated with an increased prevalence and severity of myopia, even if their children had used Ortho-K for myopia control.

Around 90% of our subjects reported good/very good unaided vision and were satisfied with Ortho-K treatment. This finding aligns with previous studies [24,25,26], which indicated that the level of satisfaction was significantly correlated with post-treatment vision. Another reason could be that Ortho-K is cosmetically desirable, allowing children spectacle-free vision. We attempted to determine which factors are related to post-wearing vision. After adjusting for other co-variates, only initial SE and regular cleaning of lens protein were significantly correlated with clear day vision. The former is to be expected because the lower target power of the Ortho-K lenses has better corneal reshaping effects and less rebound of myopia after lens removal; it has better centration of the lens than those of high-power lenses. Meanwhile, a recent study reported that tear proteins accumulated daily on the surface of Ortho-K and affected the optical characteristics of the lens [27]. The long-term deposition of tear proteins on the Ortho-K lenses could alter their reverse-geometry design and not only decrease the effectiveness of myopia control but also exacerbate the risk of infection.

Lens binding was one of the major side effects, and 34.8% of subjects had experienced it at least one time within the past 6 months. Moreover, lens decentration (15.4%) and punctate keratitis (7.9%) were also common problems. These findings corroborate previous studies [10,11,17]. Previously, several factors have been proposed to promote lens binding in Ortho-K, such as coated lens, increased viscosity of the tear film, eyelid pressure on the lens, negative hydraulic pressure in the tear film, and reverse-geometry lens designs [28]. The authors supposed that the higher tension of the corneal–lens surface could contribute to a strong suction force. However, we found that wearing 6–7 days/week was correlated with less risk of lens binding. We speculate that wearers may expertly practice with more wearing days and will not omit the proper steps. On the other hand, Maseedupally et al. [29] reported that increased corneal toricity increased amounts of treatment zone decentration; however, we found that initial astigmatism or use of toric lenses was not a risk factor for lens decentration. Regarding punctate keratitis, Gispets et al. described that the incidence of corneal staining increased with the duration of Ortho-K treatment [30]. Nevertheless, the daily wearing time, weekly wearing days, and duration of treatment were not related to punctate keratitis in the study; adequate wearing time (7–8 h) and regular lens replacement (1.5–2 years) might be the possible reasons.

Although itching, secretion, tearing, and blinking were common discomfort symptoms, their severity was graded as mild by most Ortho-K users and parents. After reviewing their medical files, allergic conjunctivitis was the most popular impression; this symptom improved in most cases after using a medication and no major complications were found in the following visits. Moreover, various problems have been posted on our social media by users and parents, including eye discomfort, lens condition, and so on. Thus, a prompt response was raised by the doctor, and a clinic visit was suggested as soon as possible if the condition persists.

Our study has some limitations. First, our sample might not be completely representative because some users with poor compliance did not return to visit regularly. Accordingly, we designed the study period for 12 months and collected suitable subjects as far as possible. Second, the questionnaire results could be biased because some questions were answered by parents, which might have influenced the actual response of the children. Third, the effect of myopia control with Ortho-K was not linked to the questionnaire results because the subjects had a different duration of wearing Ortho-K, and some of them were not followed up long enough. Nevertheless, 83% of parents were satisfied with the myopia control effect, and the effects of Ortho-K on refractive change and axial length elongation were reported in our previous studies [31]. Thus, additional prospective studies are warranted to investigate the correlation between subjective feedback and the effect of myopia control.

## 5. Conclusions

This study reports a high degree of satisfaction with Ortho-K use in a tertiary hospital in Taiwan. The most powerful motivation for parents is the rapid progression of myopia and Ortho-K treatment is mostly chosen because of its effectiveness. Time spent on near-work activities and using electronic devices increases with age among the Ortho-K wearers. In addition, clear day vision was strongly correlated with initial SE and the regular cleaning of lens protein. Wearing the Ortho-K lenses >6 days/week is accompanied by less risk of lens binding. A comprehensive care program could improve compliance among Ortho-K users. In summary, Ortho-K could be the safe and satisfying option for children to correct vision and alleviate myopia progression.

## Figures and Tables

**Figure 1 jcm-11-04126-f001:**
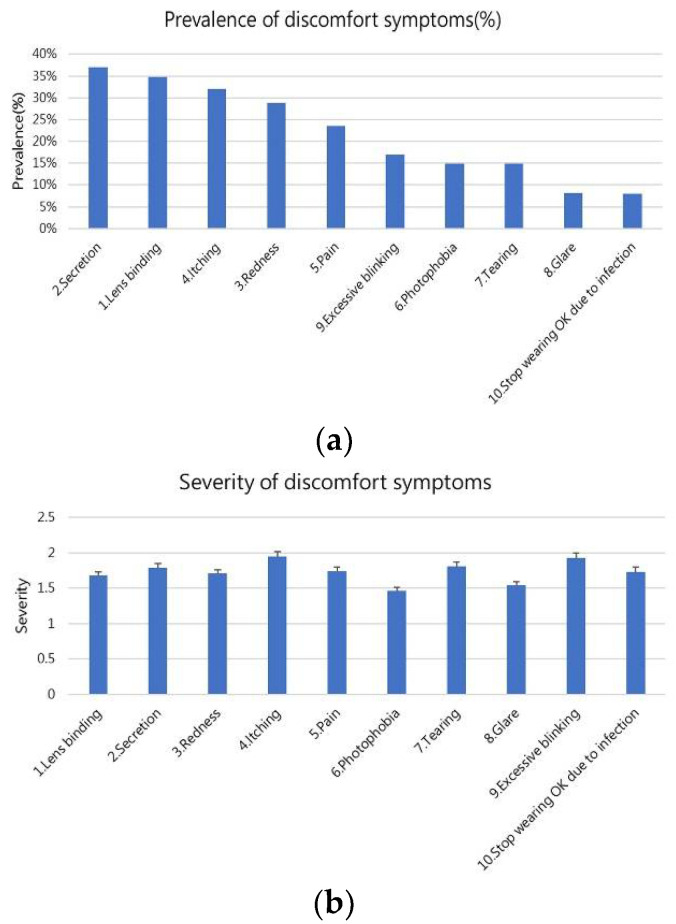
Prevalence (**a**) and severity (**b**) of discomfort symptoms.

**Figure 2 jcm-11-04126-f002:**
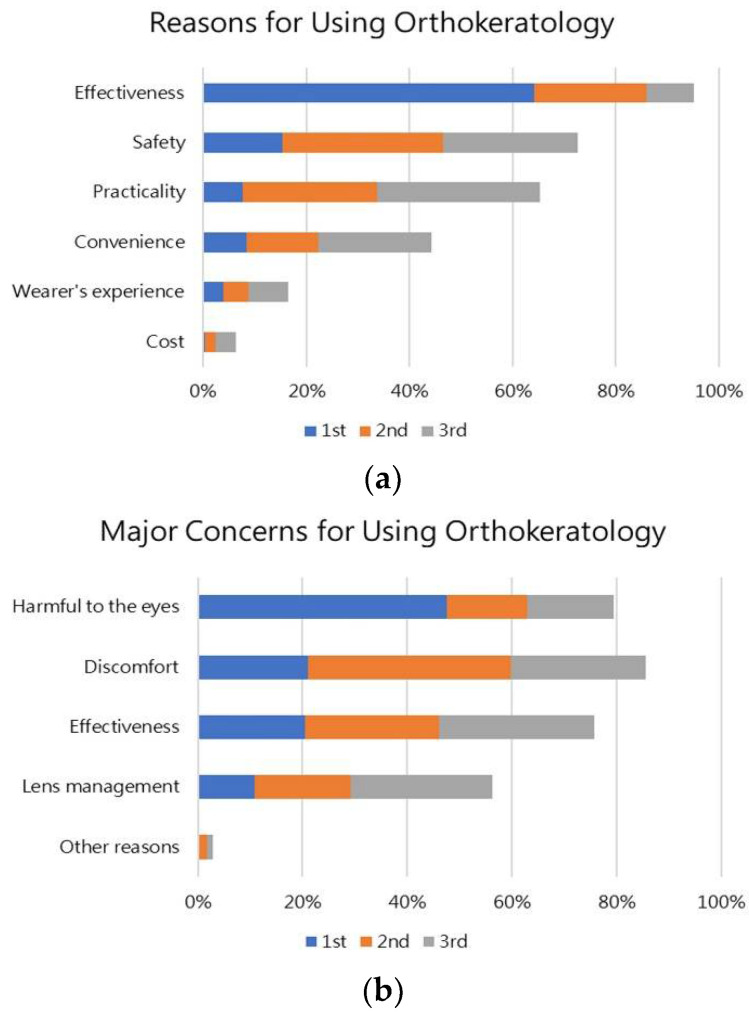
The reasons (**a**) and concerns (**b**) of Ortho-K use from the parents/guardian.

**Table 1 jcm-11-04126-t001:** Demographics of orthokeratology wearers.

Variable	Male	Female	Total	*p*-Value
Numbers	155	150	305	
**Age**	12.77 ± 2.87	13.49 ± 3.83	13.13 ± 3.39	0.066
**Toric Orthokeratology (%)**	24 (15.5%)	38 (25.3%)	62 (20.3%)	0.033 *
**Right Eye**				
Initial Visual Acuity (logMAR)	0.06 ± 0.04	0.05 ± 0.04	0.06 ± 0.04	0.235
Initial Sphere	−3.22 ± 1.81	−3.80 ± 1.88	−3.51 ± 1.86	0.006 *
Initial Cylinder	−0.83 ± 0.65	−1.05 ± 0.96	−0.94 ± 0.83	0.022 *
Average Keratometry	42.95 ± 1.20	43.51 ± 1.44	43.23 ± 1.35	<0.001 *
Initial Axial Length	25.30 ± 1.08	24.98 ± 1.07	25.14 ± 1.08	0.012 *
**Left Eye**				
Initial Visual Acuity (logMAR)	0.06 ± 0.04	0.05 ± 0.04	0.06 ± 0.04	0.531
Initial Sphere	−3.10 ± 1.84	−3.66 ± 1.85	−3.38 ± 1.86	0.009 *
Initial Cylinder	−0.98 ± 0.69	−1.19 ± 0.95	−1.08 ± 0.84	0.026 *
Average Keratometry	43.00 ± 1.22	43.53 ± 1.46	43.26 ± 1.36	0.001 *
Initial Axial Length	25.25 ± 1.09	24.95 ± 1.09	25.10 ± 1.09	0.017 *
**Father’s education level (n = 299)**				0.977
High school and lower	29 (19.3%)	29 (19.5%)	58 (19.4%)	
College and higher	121 (80.7%)	120 (80.5%)	241 (80.6%)	
**Mother’s education level (n = 299)**				0.721
High school and lower	29 (18.8%)	25 (17.2%)	54 (18.1%)	
College and higher	125 (81.2%)	120 (82.8%)	245 (81.9%)	
**Father’s refractive status (n = 299)**				0.437
Low myopia (>−3D)	72 (48.3%)	61 (41.2%)	134 (44.8%)	
Moderate myopia (between −6D and −3D)	46 (30.5%)	49 (33.1%)	95 (31.8%)	
High myopia(<−6D)	32 (21.2%)	38 (25.7%)	70 (23.4%)	
**Mother’s refractive status (n = 301)**				0.326
Low myopia (>−3 D)	52 (33.5%)	58 (39.7%)	110 (36.5%)	
Moderate myopia (between −6 D and −3D)	59 (38.1%)	44 (30.1%)	103 (34.2%)	
High myopia (<−6D)	44 (28.4%)	44 (30.1%)	88 (29.2%)	
**How did they obtain the information of Ortho-K**				
Recommendation from relatives and friends	99 (63.9%)	95 (63.3%)	194 (63.6%)	0.922
Recommendation from doctors	51 (32.9%)	58 (38.7%)	109 (35.7%)	0.294
Internet	39 (25.2%)	32 (21.3%)	71 (23.3%)	0.429
Health education instructions	6 (3.9%)	7 (4.7%)	13 (4.3%)	0.731
Myopia control conference	5 (3.2%)	3 (2.0%)	8 (2.6%)	0.723

Data are presented as n (%) or mean ± standard deviation. * *p*-value < 0.05 was considered statistically significant after test. Abbreviations: D = Diopter; Ortho-K = Orthokeratology.

**Table 2 jcm-11-04126-t002:** Characteristics of wearers in myopia control.

Variable	Male	Female	Total	*p*-Value
N	155	150	305	
**Duration after onset of myopia**				0.057
<1 year	14 (9.2%)	8 (5.3%)	22 (7.3%)	
1–3 years	55 (35.9%)	40 (26.7%)	95 (31.4%)	
≥3 years	84 (54.9%)	102 (68.0%)	186 (61.4%)	
**Previous spectacles wearing**				0.841
No	42 (30.0%)	35 (25.7%)	77 (27.9%)	
<1 year	21 (15.0%)	19 (14.0%)	40 (14.5%)	
1–3 years	34 (24.3%)	36 (26.5%)	70 (25.4%)	
≥3 years	43 (30.7%)	46 (33.8%)	89 (32.2%)	
**Previous SCL wearing**				0.071
No	110 (94.8%)	98 (84.5%)	208 (89.7%)	
<1 year	4 (3.4%)	10 (8.6%)	14 (6.0%)	
1–3 years	1 (0.9%)	3 (2.6%)	4 (1.7%)	
≥3 years	1 (0.9%)	5 (4.3%)	6 (2.6%)	
**Atropine eyedrops use**				0.322
No	27 (18.6%)	31 (22.6%)	58 (20.6%)	
<1 year	61 (42.1%)	43 (31.4%)	104 (36.9%)	
1–3 years	37 (25.5%)	40 (29.2%)	77 (27.3%)	
≥3 years	20 (13.8%)	23 (16.8%)	43 (15.2%)	
**Reasons for wearing OK**				
Inconvenient with glasses wearing	103 (66.5%)	95 (63.3%)	198 (64.9%)	0.568
Discomfort with Atropine use	20 (12.9%)	20 (13.3%)	40 (13.1%)	0.911
Recommended by others	36 (23.2%)	45 (30.0%)	81 (26.6%)	0.180
Efficacy in myopia control	121 (78.1%)	112 (74.7%)	233 (76.4%)	0.485
Only night wearing	71 (45.8%)	74 (49.3%)	145 (47.5%)	0.537
**Why choosing fitting Ortho-K in this hospital**				
Reasonable price	15 (9.7%)	18 (12.0%)	33 (10.8%)	0.514
Professionalism	83 (53.5%)	86 (57.3%)	169 (55.4%)	0.506
Recommended by relatives or friends	86 (55.5%)	75 (50.0%)	161 (52.8%)	0.338
Regular follow-up	83 (53.5%)	72 (48.0%)	155 (50.8%)	0.333
Searching in internet	9 (5.8%)	7 (4.7%)	16 (5.2%)	0.655

Data are presented as n (%) or mean ± standard deviation.

**Table 3 jcm-11-04126-t003:** Insertion related information.

Variable	<12 y/o	12–15 y/o	>15 y/o	Total	*p*-Value
N	116	114	75	305	
**Lens Wearing**					<0.001 *
Oneself	41 (35.3%)	88 (77.2%)	72 (96.0%)	201 (65.9%)	
by Others	75 (64.7%)	26 (22.8%)	3 (4.0%)	104 (34.1%)	
**Lens Cleaning**					<0.001 *
Oneself	40 (34.5%)	87 (76.3%)	71 (94.7%)	198 (64.9%)	
by Others	76 (65.5%)	27 (23.7%)	4 (5.3%)	107 (35.1%)	
**Frequency of clinic visit**					0.001 *
Regular visits (Every 2–3 months)	113 (97.4%)	101 (88.6%)	57 (77.0%)	271 (89.1%)	
Regular visits (Every 5–6 months)	1 (0.9%)	6 (5.3%)	9 (12.2%)	16 (5.3%)	
Irregular visits	2 (1.7%)	7 (6.1%)	8 (10.8%)	17 (5.6%)	
**Regular replacement of lens (n = 253)**	70 (80.5%)	88 (85.4%)	56 (88.9%)	214 (84.6%)	0.352
**Lost or damaged**	26 (22.4%)	34 (29.8%)	24 (32.4%)	84 (27.6%)	0.258

Data are presented as n (%) or mean ± standard deviation. * *p*-value < 0.05 was considered statistically significant after test.

**Table 4 jcm-11-04126-t004:** Insertion behaviors of orthokeratology wearers.

Variable	Male	Female	Total	*p*-Value
N	155	150	305	
**Lens insertion by oneself**	94 (60.6%)	107 (71.3%)	201 (65.9%)	0.049 *
**Lens cleaning by oneself**	93 (60.0%)	105 (70.0%)	198 (64.9%)	0.067
**Regularly replace the lens case and holder**	141 (91.0%)	136 (90.7%)	277 (90.8%)	0.927
**Daily insertion time (n = 304)**				0.052
Less than 6 h	6 (3.8%)	14 (9.4%)	20 (6.6%)	
More than 6 h	149 (96.2%)	135 (90.6%)	284 (93.4%)	
**Average insertion days in a week (n = 302)**				0.159
Less than or equal to 5 days	28 (18.1%)	18 (12.2%)	46 (15.2%)	
Greater than or equal to 6 days	127 (81.9%)	129 (87.8%)	256 (84.8%)	
**Cleaning methods (n = 303)**				0.124
Povidone iodine-based disinfecting solution	88 (57.1%)	72 (48.3%)	160 (52.8%)	
Rub with daily cleaner	66 (42.9%)	77 (51.7%)	143 (47.2%)	
**Regularly cleaning of lens protein (n = 253)**				0.685
Yes (Average once per two weeks)	62 (51.2%)	71 (53.8%)	133 (52.6%)	
No	59 (48.8%)	61 (46.2%)	120 (47.4%)	
**Frequency of clinic visit (n = 304)**				0.423
Regular visits (Every2- 3 months)	136 (87.7%)	135 (90.6%)	271 (89.1%)	
Irregular visits	19 (12.3%)	14 (9.4%)	33 (10.9%)	
**Regular replacement of lens (n = 253)**	116 (87.9%)	98 (81.0%)	214 (84.6%)	0.130
**Lost or damaged**	45 (29.0%)	39 (26.2%)	84 (27.6%)	0.578
**Daytime vision (n = 304)**				0.280
Clear all day	133 (85.8%)	121 (81.2%)	254 (83.6%)	
Blurred after noon	22 (14.2%)	28 (18.8%)	50 (16.4%)	

Data are presented as n (%) or mean ± standard deviation. * *p*-value < 0.05 was considered statistically significant after test.

**Table 5 jcm-11-04126-t005:** Time in near work and outdoor activity (mean hours per day).

Variable	Total	<12 y/o		12–15 y/o		>15 y/o		*p*-Value
		Male	Female	Male	Female	Male	Female	
N	305	62	55	62	51	30	45	
**Time in near work during weekdays**								0.006 *
Less than 8 h	159 (52.5%)	41 (66.1%)	34 (64.1%)	25 (40.3%)	24 (47.1%)	18 (60.0%)	17 (37.8%)	
More than 8 h	144 (47.5%)	21 (33.9%)	19 (35.8%)	37 (59.7%)	27 (52.9%)	12 (40.0%)	28 (62.2%)	
**Time in near work during weekends**								<0.001 *
Less than 8 h	232 (76.6%)	55 (88.7%)	50 (90.9%)	42 (67.7%)	40 (78.4%)	18 (62.91%)	27 (61.4%)	
More than 8 h	71 (23.4%)	7 (11.3%)	5 (9.1%)	20 (32.3%)	11 (21.6%)	11 (37.9%)	17 (38.6%)	
**Time in using of electronic devices during weekdays**								<0.001 *
Less than 1 h	141 (46.4%)	49 (79.0%)	42 (76.4%)	25 (40.3%)	18 (35.3%)	4 (13.3%)	3 (6.8%)	
Between 1 to 6 h	149 (49.0%)	13 (21.0%)	12 (21.8%)	35 (56.5%)	32 (62.7%)	20 (66.7%)	37 (84.1%)	
More than 6 h	14 (4.6%)	0 (0.0%)	1 (1.8%)	2 (3.2%)	1 (2.0%)	6 (20.0%)	4 (9.1%)	
**Time in using electronic devices during weekends**								<0.001 *
Less than 1 h	62 (20.3%)	26 (41.9%)	19 (34.5%)	8 (12.9%)	6 (11.8%)	1 (3.3%)	2 (4.4%)	
Between 1 to 6 h	204 (66.9%)	33 (53.2%)	32 (58.2%)	45 (72.6%)	41 (80.4%)	20 (66.7%)	33 (73.3%)	
More than 6 h	39 (12.8%)	3 (4.8%)	4 (7.3%)	9 (14.5%)	4 (7.8%)	9 (30.0%)	10 (22.2%)	
**Time in outdoor activities during weekdays**								0.001 *
Less than 1 h	139 (45.6%)	19 (30.6%)	25 (45.5%)	34 (54.8%)	30 (58.8%)	6 (20.0%)	25 (55.6%)	
Between 1 to 6 h	159 (52.1%)	42 (67.7%)	29 (52.7%)	24 (38.7%)	20 (39.2%)	24 (80.0%)	20 (44.4%)	
More than 6 h	7 (2.3%)	1 (1.6%)	1 (1.8%)	4 (6.5%)	1 (2.0%)	0 (0.0%)	0 (0.0%)	
**Time in outdoor activities during weekends**								<0.001 *
Less than 1 h	137 (44.9%)	18 (29.0%)	24 (43.6%)	23 (45.8%)	36 (70.6%)	5 (16.7%)	31 (68.9%)	
Between 1 to 6 h	162 (53.1%)	42 (67.7%)	31 (56.4%)	36 (50.0%)	14 (27.5%)	25 (83.3%)	14 (31.1%)	
More than 6 h	6 (2.0%)	2 (3.2%)	0 (0.0%)	3 (4.2%)	1 (2.0%)	0 (0.0%)	0 (0.0%)	

Data are presented as n (%) or mean ± standard deviation. * *p*-value < 0.05 was considered statistically significant after test.

**Table 6 jcm-11-04126-t006:** Satisfaction with orthokeratology wearing.

Variable	Male	Female	Total	*p*-Value
N	155	150	305	
**Satisfaction from myopia (AL) control effect (n = 304)**				0.034 *
Acceptable	20 (12.9%)	33 (22.1%)	53 (17.4%)	
Satisfied or very satisfied	135 (87.1%)	116 (77.9%)	251 (82.6%)	
**Ease of wearing (n = 304)**				0.995
Acceptable	53 (34.2%)	51 (34.2%)	104 (34.2%)	
Easy or very easy	102 (65.8%)	98 (65.8%)	200 (65.8%)	
**Reasonable price (n = 302)**				0.392
Expensive	104 (67.5%)	93 (62.8%)	197 (65.2%)	
Acceptable	50 (32.5%)	55 (37.2%)	105 (34.8%)	
**Overall satisfaction**				0.527
Acceptable	17 (11.0%)	20 (13.3%)	37 (12.1%)	
Satisfied or very satisfied	138 (89.0%)	130 (86.7%)	268 (87.9%)	
**Willing to continue wearing**	146 (97.3%)	146 (99.3%)	292 (98.3%)	0.371

Data are presented as n (%) or mean ± standard deviation. * *p*-value < 0.05 was considered statistically significant after test. Abbreviations: AL = Axial Length.

**Table 7 jcm-11-04126-t007:** Factors associated with clear day vision and side effects.

Item	Clear Day Vision		Lens Binding		Punctate Keratitis		Decentration	
	OR (95% CI)	*p*-Value	OR (95% CI)	*p*-Value	OR (95% CI)	*p*-Value	OR (95% CI)	*p*-Value
**Age group**								
<12 years old	Ref.		Ref.		Ref.		Ref.	
12–15 years old	0.93 (0.32, 2.72)	0.891	1.26 (0.59, 2.67)	0.549	0.94 (0.27, 3.27)	0.926	0.84 (0.32, 2.22)	0.724
>15 years old	0.70 (0.18, 2.66)	0.598	0.92 (0.33, 2.56)	0.878	0.71 (0.13, 3.92)	0.697	0.78 (0.20, 3.00)	0.717
**Sex**								
Male	Ref.		Ref.		Ref.		Ref.	
Female	0.75 (0.32, 1.72)	0.493	1.62 (0.89, 2.98)	0.117	1.63 (0.58, 4.61)	0.353	1.09 (0.49, 2.42)	0.825
**Initial Spherical Equivalent**	1.75 (1.35, 2.25)	** *<0.001 ** **	0.93 (0.77, 1.11)	0.408	0.85 (0.63, 1.15)	0.298	0.97 (0.77, 1.24)	0.824
**Initial Keratometry**								
Normal	Ref.		Ref.		Ref.		Ref.	
Flatter	0.76 (0.16, 3.57)	0.723	2.53 (0.7, 9.19)	0.157	3.6 (0.59, 22.03)	0.165	0.74 (0.12, 4.53)	0.744
Steeper	3.49 (0.68, 17.98)	0.135	1.26 (0.53, 2.99)	0.605	1.79 (0.49, 6.50)	0.376	1.47 (0.52, 4.21)	0.470
**Initial Cylinder**	0.59 (0.34, 1.01)	0.056	1 (0.66, 1.51)	0.999	1.5 (0.66, 3.38)	0.330	0.77 (0.47, 1.26)	0.295
**Insertion by whom**								
Self	Ref.		Ref.		Ref.		Ref.	
Others	0.89 (0.32, 2.45)	0.817	1.38 (0.68, 2.8)	0.372	1.01 (0.32, 3.21)	0.981	0.64 (0.25, 1.61)	0.340
**Wearing time**								
<6 h	Ref.		Ref.		Ref.		Ref.	
6–8 h	1.67 (0.41, 6.85)	0.477	1.12 (0.35, 3.66)	0.846	1.81 (0.2, 16.51)	0.597	1.13 (0.22, 5.85)	0.881
>8 h	0.49 (0.10, 2.33)	0.370	0.76 (0.21, 2.77)	0.678	1.05 (0.1, 11.51)	0.968	1.89 (0.33, 10.95)	0.476
**Insertion days**								
≦5 days	Ref.		Ref.		Ref.		Ref.	
≧6 days	1.34 (0.47, 3.88)	0.584	0.46 (0.21, 0.99)	** *0.046 ** **	0.57 (0.16, 2.05)	0.393	2.04 (0.56, 7.39)	0.280
**Regularly cleaning of lens protein**								
No	Ref.		Ref.		Ref.		Ref.	
Yes	2.65 (1.08, 6.53)	** *0.034 ** **	1.09 (0.59, 2.01)	0.780	0.66 (0.24, 1.84)	0.428	2.13 (0.92, 4.93)	0.079
**Cleaning methods**								
Povidone iodine-based disinfecting solution	Ref.		Ref.		Ref.		Ref.	
Rub with daily cleaner	1.21 (0.45, 3.26)	0.700	1.00 (0.51, 1.97)	0.991	1.64 (0.52, 5.15)	0.395	0.79 (0.32, 1.94)	0.608
**Frequency of clinic visit**								
Non-regular visits	Ref.		Ref.		Ref.		Ref.	
Regular visits (Every 2–3 months)	2.42 (0.81, 7.25)	0.113	0.95 (0.37, 2.43)	0.913	3.33 (0.38, 29.39)	0.279	2.23 (0.47, 10.56)	0.313
**Duration of wearing**	0.97 (0.92, 1.03)	0.330	0.98 (0.94, 1.02)	0.238	0.97 (0.91, 1.04)	0.410	1.02 (0.97, 1.08)	0.341
**Toric orthokeratology**								
No	Ref.		Ref.		Ref.		Ref.	
Yes	0.49 (0.17, 1.4)	0.181	1.49 (0.67, 3.32)	0.327	0.92 (0.22, 3.81)	0.912	0.78 (0.28, 2.24)	0.651

Data are presented as Odds ratio (95% CI), * *p*-value < 0.05 was considered statistically significant after test. Abbreviations: Ref. = Reference.

## Data Availability

The data presented in this study are available on request from the corresponding author. The data are not publicly available due to patients’ confidentiality.

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
