# Peer review of "Assessment of Satisfaction, Compliance and Side Effects among Long-Term Orthokeratology Wearers"

_jcm, 2022, doi:10.3390/jcm11144126_

Round 1

Reviewer 1 Report

Please view the attachment

Reviewer 2 Report

It seems that there was a great effort from the authors to compile so much information in a single paper. The work is original and of interest to the orthoK practioner, as well as may give more reassurance and confidence to practitioners less comfortable with treatment. The work is well structured and easy to read.

Tables are long, and would probably be presented in another way, but the choice for tables is understandable.

The scales of the graphs should be standardized (e.g. number of decimal places), and when the decimal numbering of the entire scale are only zeros, the values should be shown without decimal numbering.

Reviewer 3 Report

The authors aim to assess the compliance. Daily activities, and satisfaction among the long term ortho-k users in a tertiary hospital in Taiwan and analyze the side effects and related risk factors and they found that the main reasons for using ortho-k were effectiveness, safety and practically, while the major concerns were discomfort, harmful to the eyes and no effect.

The introduction is exceptionally long and diffuse, orthokeratology starts at the middle section of the introduction and the information provided was very general and do not focus on the topic of the manuscript.

The manuscript is extremely limited from the point of scientific soundness since only a questionnaire was provided to the subjects. No clinical or in-vivo findings were evaluated and compared with the questionnaire described. If only the information given for the patients were collected, no factual issues were tested or demonstrated. Just the visual acuity and slit lamp findings is not enough to establish a correlation study between the questionnaires and the clinical observations.

The results only represent a descriptive analysis only represent the difference between two types of population and there was not enough data to defeat this results.

Findings do not support the discussion section or the conclusion

Round 2

Reviewer 3 Report

Limitation of the manuscript are still present

The reject decision is clear

Author Response

We feel sorry that our manusript does not meet your expectation. Anyway, we appriciate you taking time to review our manuscript.